# Research and application of OPC UA-based digital twin smoke alarm calibration workshop information model

Min Wu[1]*, Wenfeng Ying[2]

1 School of Mechanical and Electrical Engineering, Ningbo Polytechnic, Ningbo, Zhejiang, China, 2 Ningbo Weizmat Electronics Co., Ltd., Ningbo, Zhejiang, China

* 53760012@qq.com

**Data Availability Statement:** All relevant data are within the manuscript and its Supporting Information files.

**Funding:** This work was supported by 2022 Visiting Engineer of Colleges and Universities in

## Abstract

Aiming at the problems of data integration and information sharing in the calibration workshop of smoke alarm, combining OPC Unified Architecture (OPC UA) and digital twin technology, the OPC UA based digital twin smoke alarm calibration workshop information network framework is proposed. Using OPC UA information modeling technology, a digital workshop information model for smoke alarm calibration was constructed, and the static attribute set, process attribute set, and functional component set of the workshop were defined in detail. Taking the smoke box as an example, the information model construction was instantiated, and data collection and transmission were carried out with the digital twin model of the smoke box, achieving the implementation mapping between the twin model and physical entities. The results show that based on the proposed OPC UA information model, the interconnection and intercommunication of data at all levels of the digital twin smoke alarm calibration workshop can be achieved, and the workshop can be visualized, digitized, and flexible.

## 1. Introduction

With the successive propositions of strategies such as Industry 4.0 and Made in China 2025, intelligent manufacturing has become the core of development for countries at the advent of the fourth industrial revolution. As the best means of integrating interactive physical world and virtual world, digital twin is a major driving force to promote the further transformation of manufacturing workshop towards intelligence, personalization and service [1]. With the increasing demand for smoke alarms, their quantity and variety are also increasing. The calibration workshop of smoke alarms will generate massive data during production and operation. How to use these data to establish a multi-domain integrated data model and achieve an efficient application model of real-time interaction between data model and physical entity is a problem to be solved in the production process of smoke alarms [2]. Digital twin technology provides a possibility to solve this problem [3].

The core of digital twinning is modeling and data interaction [4]. Designing a reasonable information-physical fusion system for digital twinning workshop is the foundation for

Zhejiang Province(Project No. FG2022037). The funders had no role in study design, data collection and analysis, decision to publish, or preparation of the manuscript.

building virtual-real interaction in digital workshop [5], and it is also the key to solve the problem of information island in digital workshop. OPC Unified Architecture (OPC UA) protocol is one of the current common data transmission standards in the industrial field, which realizes the semantic unification of data between various layers in intelligent manufacturing workshop [6]. Regarding the application of OPC UA technology, Song, et al. [7] implemented data integration and unified management of intelligent workshop equipment information based on OPC UA technology. Li, et al. [8] proposed a textile intelligent dyeing and finishing workshop information model scheme based on OPC UA technology, which realized data communication between equipment end and manufacturing execution system. Wang, et al. [9] proposed a worksheet definition format (WDF) file model based on OPC UA, which can realize open sharing of information interaction between various levels in digital manufacturing workshop of CNC machine tools and solve the problem of information heterogeneity. Zhou, et al. [10] designed a network acquisition server based on OPC UA standard architecture to solve the networking problem of various CNC equipment. Chen, et al. [11] implemented three-dimensional real-time monitoring of digital twinning workshop operation based on multi-source heterogeneous data acquisition and transmission mode in workshop based on OPC UA technology.

The application research of OPC UA in smoke alarm related fields was conducted by searching keywords "smoke alarm," "smoke detector," or "smoke sensor" in CNKI, Wanfang Data, Web of Science, Scopus, Baidu Library and other databases. The retrieved articles mainly focused on smoke alarm design, smoke alarm detection system design, and smoke alarm monitoring technology. However, there is a lack of scholarly research on the production aspect of smoke alarms. A search using the keywords "smoke alarm and OPC UA" or "smoke detector and OPC UA" or "smoke sensor and OPC UA" yielded no results. It is evident that there is a gap in data information related to the production workshop of smoke alarms. Therefore, it is crucial to optimize the production process and strengthen workshop management research for smoke alarm manufacturing enterprises.

In summary, this article studies the information model of the digital twin smoke alarm calibration workshop based on OPC UA, constructs an OPC UA information interaction network architecture for the hoist system based on four types of multi-source and multi-dimensional heterogeneous data including people, machines, objects, and environment, analyzes the functional models and information flows that meet the requirements of the smoke alarm calibration workshop, and proposes an information modeling method applicable to the smoke alarm calibration workshop. The feasibility of the information model of the smoke alarm calibration workshop is verified through the built digital twin system platform.

## 2 OPC UA information modeling technology

The OPC Unified Architecture provides a standardized, synchronous or asynchronous, and distributed communication mechanism [12]. Under this mechanism, it allows access to different types of data horizontally or vertically. OPC UA components can be combined in different forms, on different platforms, and by different vendors [13]. Their scale can range from embedded OPC UA components in devices or controllers, to machines or complete sets of equipments using network management to provide OPC UA functionality, all the way to OPC UA server clusters [14]. OPC UA is generally based on a distributed client-server model [15].

### 2.1 OPC UA server

According to the functional requirements of the digital twin smoke alarm calibration workshop, an aggregated soft OPC UA server architecture is adopted, as shown in Fig 1. The

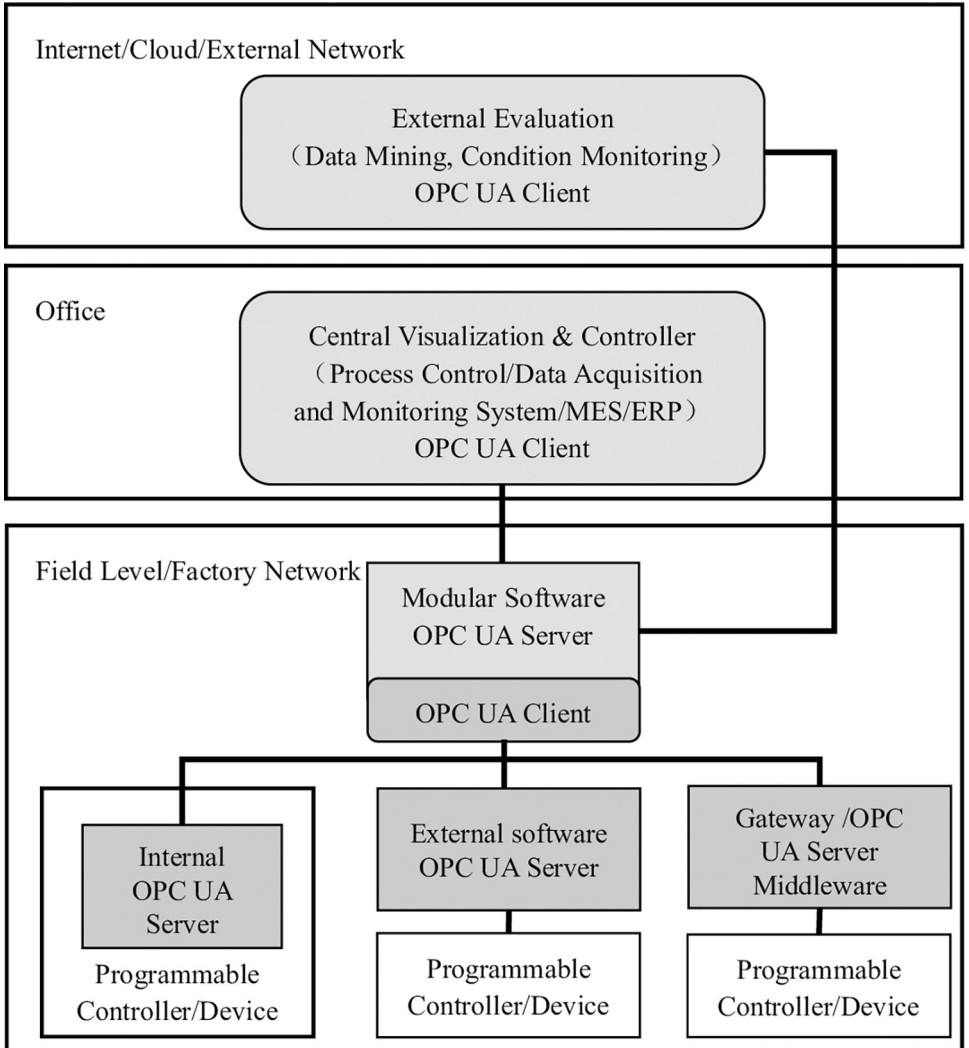

**Fig 1. Aggregated Soft OPC UA architecture.**

aggregated OPC UA server unifies the OPC UA servers for each asset, incorporates the information models of all servers in the lower layer, and integrates them at the system aggregation level [16]. This architecture type can be connected to enterprise MES or ERP, or used to separate manufacturing systems from later data analysis or earlier planning [17]. The OPC UA client can be either a SCADA, MES, ERP, or central control system maintained by the IT department, a visualization terminal running near the machine, or even an external application [18].

## 2.2 OPC UA address space and information model

The object model of OPC UA allows the integration of data, alarms, events, and historical data into an OPC UA server address space. The objects and related information provided by the OPC UA server to the client are called the address space. The address space of the OPC UA server is a fully interconnected information model that is presented in a graph-like topology [19]. This information model includes nodes, the characteristics of the nodes themselves, and the interconnections between nodes. OPC UA defines eight node classes, each with a defined

set of properties. Any node in OPC UA can be connected to each other. The OPC UA server provides the client with object type definitions that are accessed from the address space, and an information model to describe standardized nodes in the server address space.

### 2.3 OPC UA application architecture

Fig 2 shows the application architecture of OPC UA client and server. OPC UA uses client/ server mode to implement information exchange. The client application uses OPC UA client API to send and receive service requests and responses from OPC UA server. The server

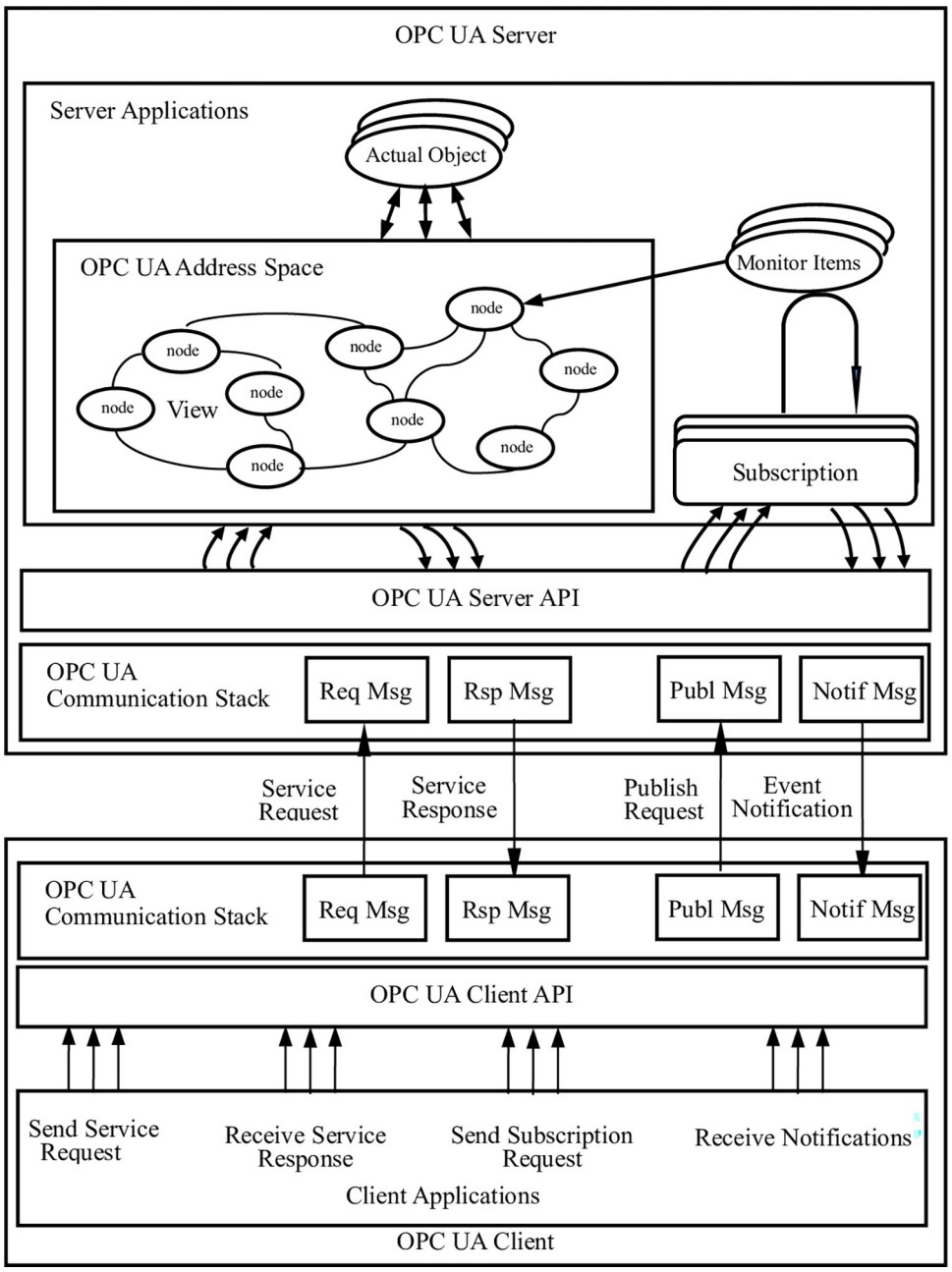

**Fig 2. OPC UA client and server application architecture.**

application uses OPC UA server API to send and receive messages from OPC UA client. Clients can access nodes in the server using OPC UA services [20].

## 3 Digital twin smoke alarm calibration workshop information model

### 3.1 Smoke alarm calibration workshop architecture

The smoke alarm calibration workshop is structured with a management layer, an execution layer, and a base layer, as illustrated in Fig 3. The management layer primarily comprises enterprise resource planning systems and product life cycle management systems to address workshop management issues. The execution layer encompasses production process management, production process quality management, production scheduling management, equipment management, order management, logistics management, and manufacturing operation management to oversee various business activities and related assets in the production process. The base layer consists of AGV cars, smoke boxes, industrial robots, intelligent printers, and intelligent workstations.

According to GB/T7393-2019 "General Technology of Digital Workshop", the essential functionalities of a digital workshop should encompass workshop planning and scheduling, production logistics management, process execution and management, workshop equipment management, and production process quality control. In conjunction with the requirements of smoke alarm calibration process, the data flow between functional modules of smoke alarm calibration workshop is illustrated in Fig 4.

According to the organizational structure, functional modules and data flow requirements of the smoke alarm calibration workshop, the physical equipment and calibration process data were digitized and modeled. They were then organized according to information modeling rules to form a digital smoke alarm workshop information model. The hierarchical architecture of the workshop information model is illustrated in Fig 5. The workshop information model corresponds to the entire workshop, while the functional component set corresponds to each functional module of the workshop, and the resource component set represents all types

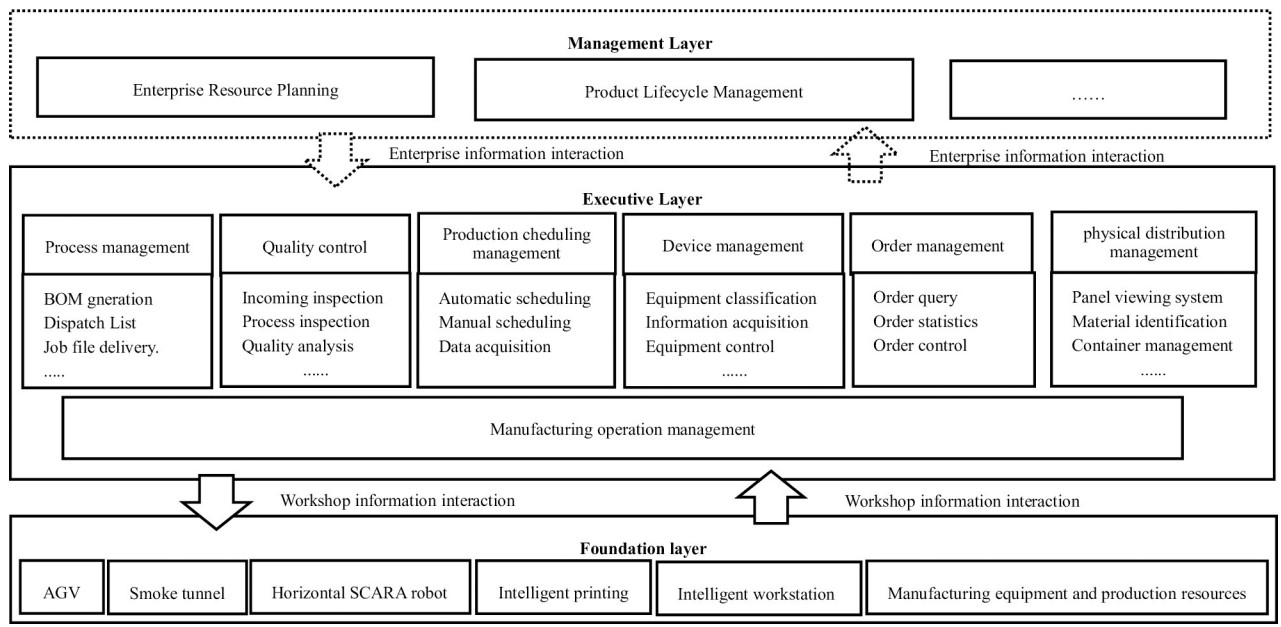

**Fig 3. Architecture of smoke alarm calibration digital workshop [21].**

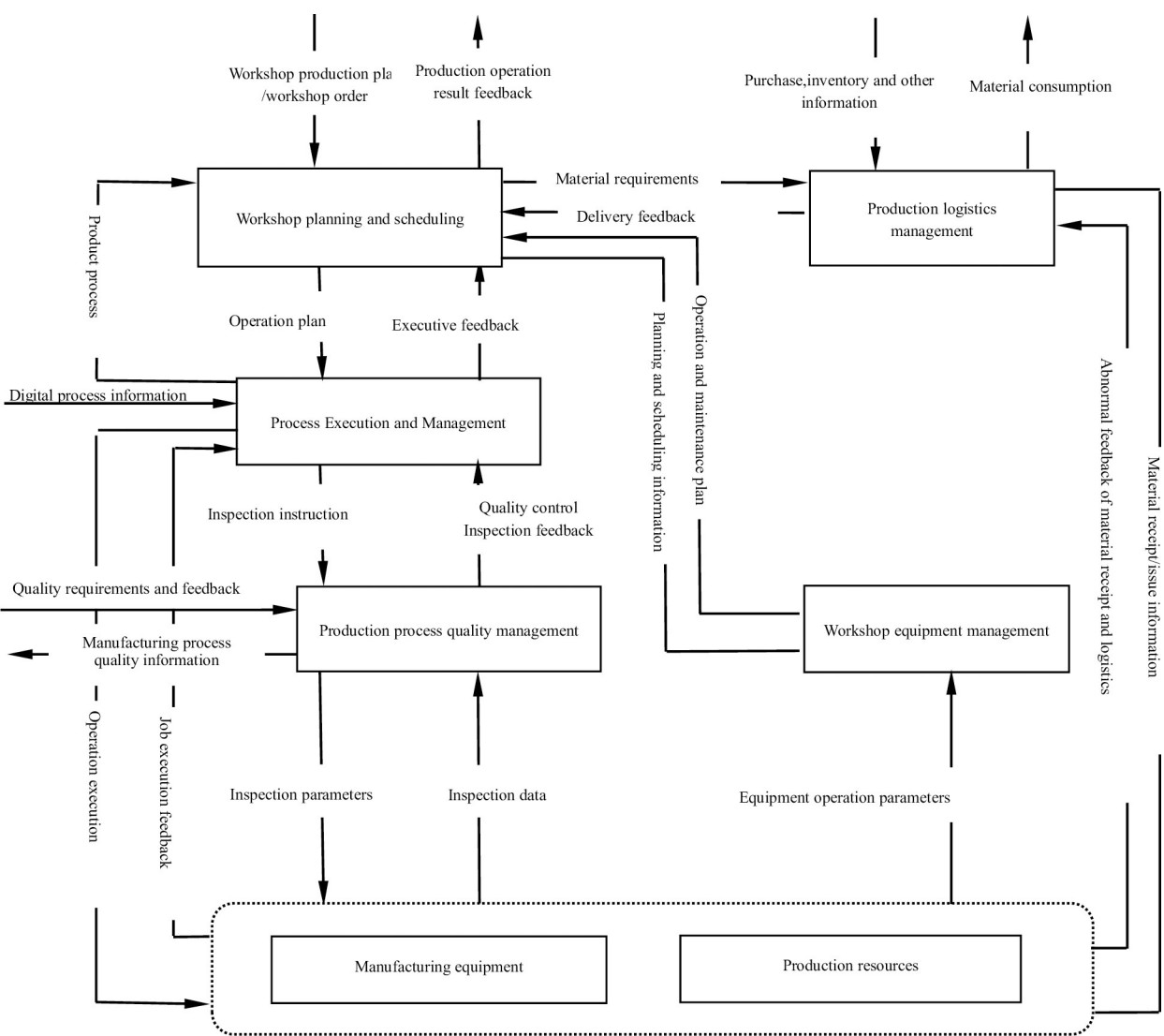

**Fig 4. Digital workshop data flow diagram [21].**

of resources involved in each module. The attribute set represents a collection of attributes with a fixed structure that describes the correlation between devices, components, attribute sets, and attributes [22].

## 3.2 Digital twin smoke alarm calibration workshop information network architecture

The data integration and fusion of multi-source heterogeneous data in the digital twin workshop are the connection channels to realize the control and management of the production workshop and to open up the interactive mapping between the virtual and real workshop. Based on the operation mode of the digital twin workshop and the requirements of virtual and real integration, an OPC UA-based digital twin smoke alarm calibration workshop information network architecture is proposed. According to the functional requirements of the workshop, the system architecture is divided into three layers: application layer, processing layer, and perception layer, as shown in Fig 6.

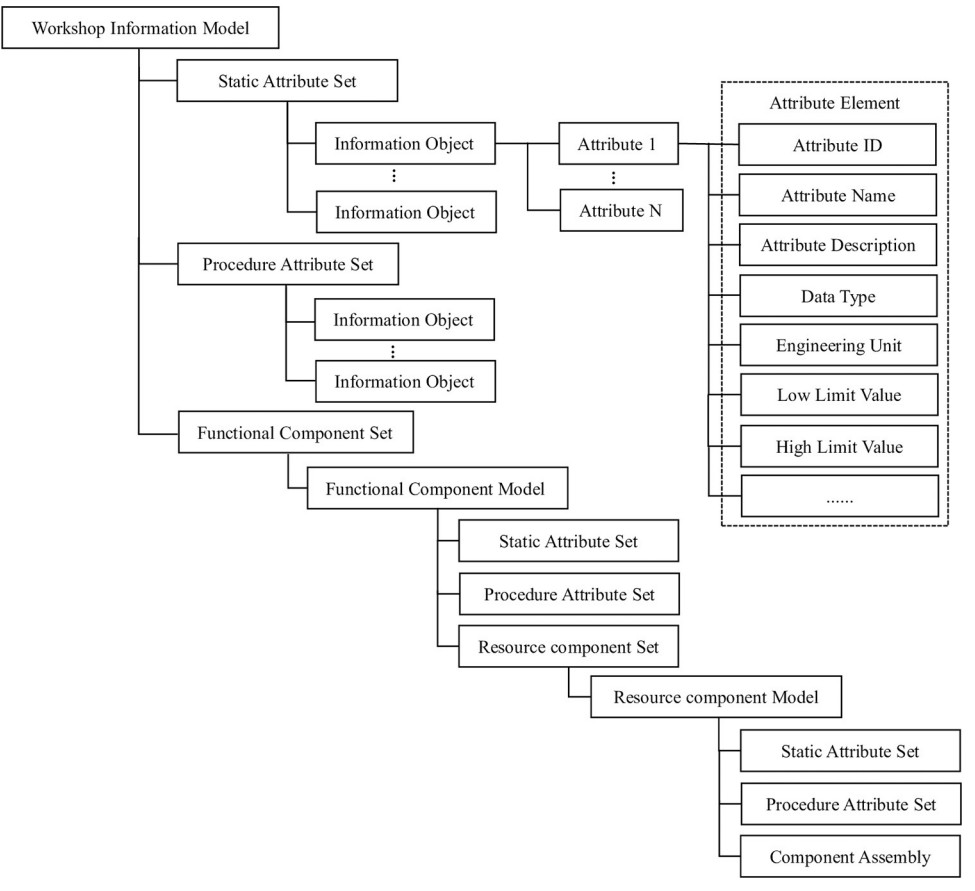

**Fig 5. Workshop information model hierarchy.**

The perception layer consists of PLC, smoke box, various sensors, industrial robots, AGV, RFID and other devices. These devices are used to collect workshop data. Most of the on-site perception devices have integrated OPC UV servers internally, which can be connected to the OPC UV server through internal communication. For devices without built-in OPC UV servers, communication with the OPC UV server can be achieved through gateways, middleware, routers, etc.

The processing layer establishes a standardized information model for multi-source heterogeneous data generated by different devices in the sensing layer through the OPC UV communication protocol. Based on the characteristics of OPC UA modeling and transmission, the standardized information model is combined with the address space module and the collected normalized data to construct the address space of the OPC UA server, and the information model is instantiated. Then, through the OPC UA server, the address space data mapping between the physical model and the twin model is completed, and the OPC UA client is directly integrated into the digital twin service platform, providing a unified communication architecture for digital twin service and data collection.

The application layer is the application program that obtains the address space of the aggregated soft OPC UA server. The OPC UA client application communicates with the aggregated soft OPC UA server in the processing layer through the OPC UA protocol, including digital twin software, workshop twin database, information system and workshop service system. The digital twin software provides a simulation environment and uses OPC UA communication to

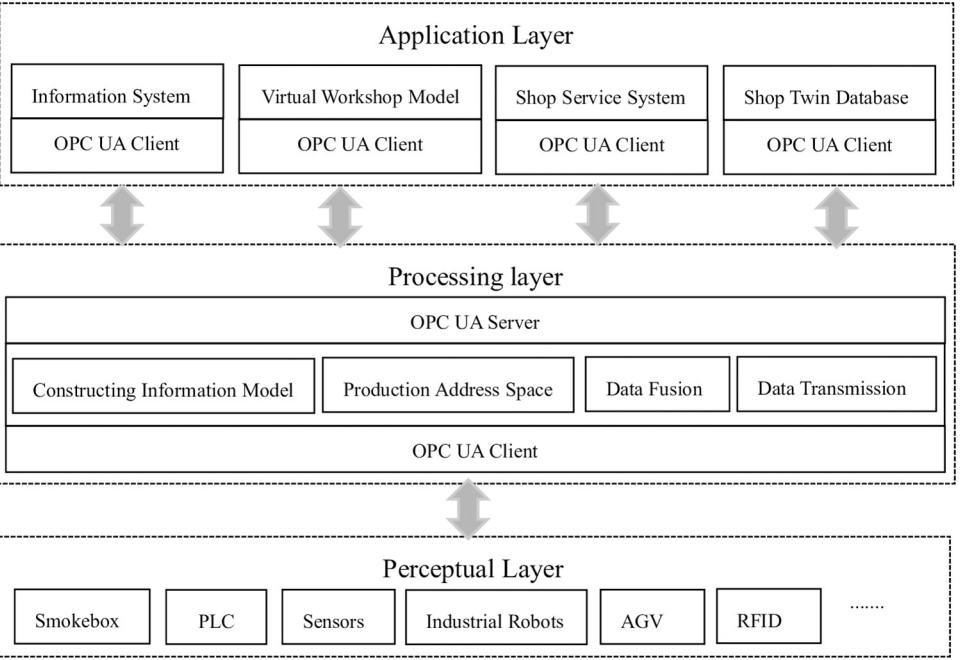

**Fig 6. Information network architecture of digital twin smoke alarm calibration workshop.**

achieve virtual-real mapping. The information system obtains data information and provides data analysis and production plan control and adjustment.

### 3.3 Digital twin smoke alarm calibration shop information model

Refer to GB/T 38869–2020 OPC UA based Digital Workshop Interconnection Network Architecture and GB/T 37928–2019 Digital Workshop Machine Tool Manufacturing Information Model, sort out the integrated information of the smoke alarm calibration workshop, and use OPC UA modeling method to obtain the smoke alarm calibration workshop information model, as shown in Fig 7.

The information model for the smoke alarm calibration workshop mainly includes static attribute sets, process attribute sets, and functional component sets. The functional component sets include four sub-models: production operation management information model, maintenance operation management information model, quality operation management information model, and logistics operation management information model [23]. Each sub-model includes its own static attribute sets, process attribute sets, and resource component sets.

The smoke alarm calibration workshop static attribute set defines the basic static information of the workshop operation, including the basic information of the workshop (such as the name of the workshop, the location of the workshop, the area of the workshop, the person in charge of the workshop, etc.), production organization (such as the departments and work groups in the workshop), work calendar (such as the working days and rest days of the workshop), and smoke alarm calibration work orders (order information obtained by the MES system from the ERP system). The process attribute set defines the dynamic data of the workshop process, which is generally summarized statistical data and order tracking information of various aspects of the workshop, and is used to provide feedback to the ERP system, such as order tracking, calibration statistics, quality statistics, inventory statistics, maintenance statistics, etc. [24].

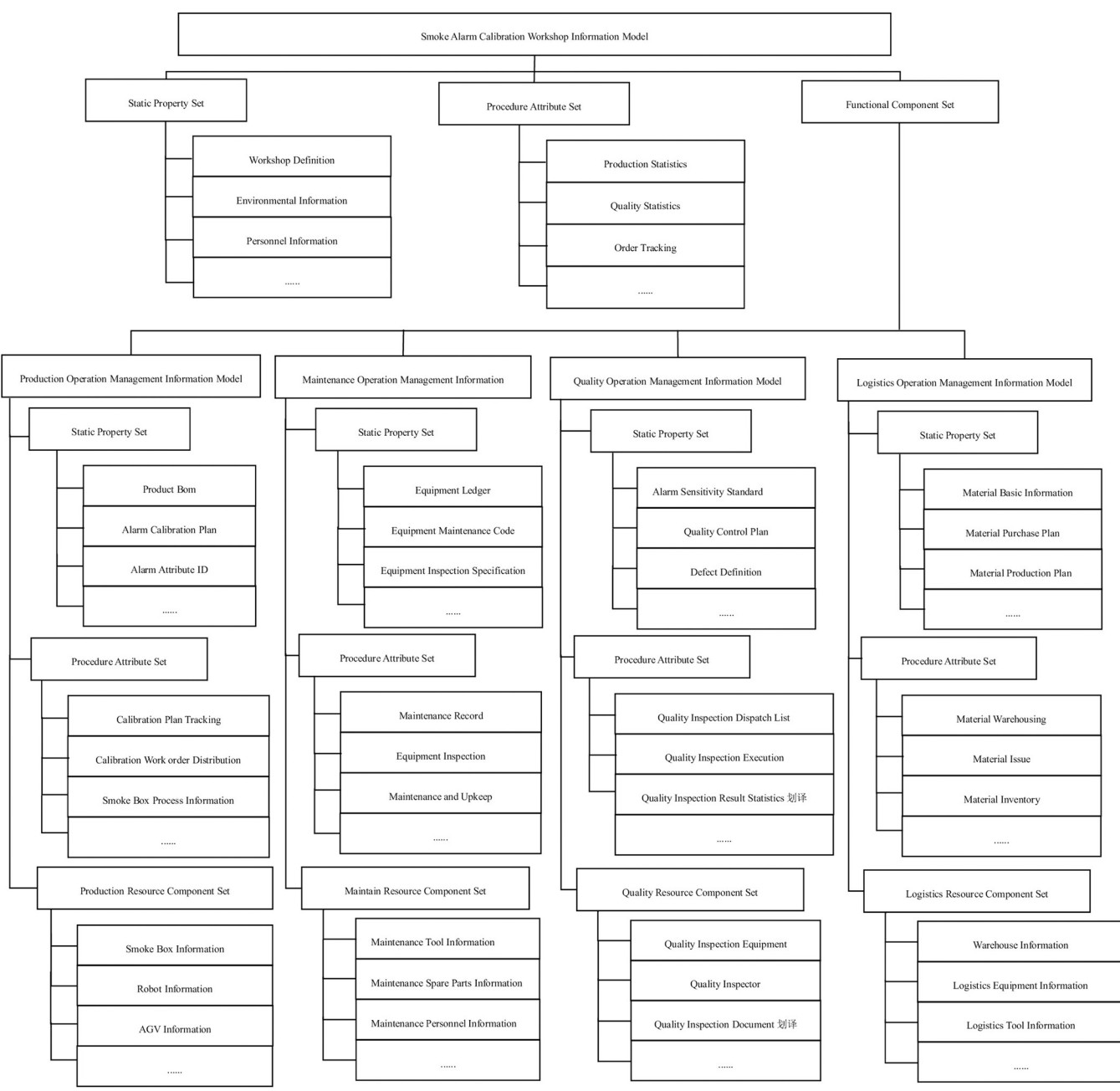

**Fig 7. Information model of smoke alarm calibration workshop.**

The static attribute set of each sub-model in the functional component set defines the set of static attribute data related to the implementation of each functional module in the smoke alarm calibration workshop. The static attribute data refers to data that does not change or changes slowly over a certain period of time. The process attribute set defines the data collected during the calibration process that may change as the calibration process progresses. The resource component set defines the information model of the equipment, tooling and personnel that make up the smoke alarm workshop.

The production operation management information model process attribute set defines the information objects and their data attributes generated during the smoke alarm calibration process, including calibration plan tracking, calibration work order dispatch, smoke box operation information, and actual job information. The production resource component set of the production operation management information model defines the information models related to equipment, tooling, personnel, etc. for smoke alarm calibration, including smoke boxes, robots, AGV cars, calibration personnel, etc.

The static attribute set of the smoke alarm calibration workshop maintenance operation management information model defines the static information set involved in daily management and maintenance of equipment, including equipment account, equipment maintenance specifications, equipment point inspection specifications, etc. The process attribute set of the maintenance operation management information model defines the information objects and their attribute sets generated during the maintenance process of equipment, including equipment repair, equipment maintenance, equipment point inspection, equipment failure, equipment alarm, equipment performance, etc. The maintenance resource component set includes maintenance tools, maintenance equipment, and maintenance personnel information.

The static attribute set of the smoke alarm calibration workshop quality operation management information model defines the static information set involved in the quality inspection of the alarm, including various alarm sensitivity standards, quality inspection plans, and defect definitions. The process attribute set of the quality operation management information model defines the information objects and their attribute sets generated by the execution of the quality inspection process and its results, including quality inspection dispatch orders, quality inspection execution, quality inspection certificates, non-conforming product reports, rework and repair orders, quality inspection plan tracking, and scrap product processing. The resource component set of the quality operation management information model includes information such as quality inspection personnel, quality inspection equipment, and quality inspection tools.

The static attribute set of the smoke alarm calibration workshop logistics operation management information model defines the static information set related to logistics management, outbound and inbound warehousing, and distribution and shipping during the operation process of the workshop, including material basic information, material procurement plan, material production plan, etc. The process attribute set of the logistics operation management information model defines the information model objects and their attribute sets generated by dynamic behaviors such as material allocation and inbound and outbound warehousing involved in the calibration process of the alarm, including material outbound, material inbound, material transfer, material inventory, material return, material distribution, etc. The resource component set of the logistics operation management information model includes warehouse information model, logistics equipment information model, logistics tool information model, etc.

## 4 Information model instantiation and application

The smoke alarm calibration workshop information model mentioned above serves as an abstract framework. In the process of digitally modeling the actual workshop, various elements within this framework must be populated according to their respective categories and semantics, based on the specific functions and equipment of the workshop, in order to create information model objects with practical significance. This process is known as instantiation of information model.

The smoke alarm calibration digital workshop of Ningbo Witzmat Electronics Co., Ltd. was chosen as the model for the information system and its application verification. The workshop

consists of 10 sets of automatic smoke boxes, 10 sets of automatic labeling machines, 10 sets of automatic coding machines, a horizontal SCARA robot production line, and various auxiliary equipment. The production line is equipped with advanced intelligent devices for automated storage, loading/unloading, detection, labeling, and coding. It is also integrated with product life cycle management, enterprise resource planning systems, and production line control software to achieve automated storage, loading/unloading, detection/calibration operations along with barcode tracking for automated production logistics [21].

The process flow for calibrating the smoke alarm is: using an AGV car for the product— workshop material warehouse—feeding—conveying (distributing)—horizontal SCARA robot picking up material—recording coordinates with a camera—placing the product in the fixture position—entering the smoke box—automatically setting the PPM value of the smoke box— communicating with the product—telling the product the current PPM value and calibrating it—exiting the warehouse—horizontal SCARA robot picking up material—reading the QR code—recording—placing the finished product in the channel—conveying—workshop material warehouse.

## 4.1 Modeling of the smoke box information model

Due to the complexity of the information model for the calibration workshop of smoke alarms, the smoke box is the most important device for calibrating the workshop. This article focuses on the instantiation of the information model with the smoke box as the object. The standard smoke box consists of several components, physical attributes, and various operations. Each component contains other subcomponents and physical attributes, so it is necessary to define relevant information model elements to abstract and describe the smoke box. Fig 8 shows the mapping between the smoke box and information model elements.

The information model of the smoke box is shown in Fig 9. The standard smoke box information model consists of a static attribute set, a process attribute set and a component assembly set. The component assembly set mainly consists of four subcomponents: a transmission device, a smoke generating device, an airflow regulating device, and a measuring device. The subcomponent information model includes static and process attributes. The static

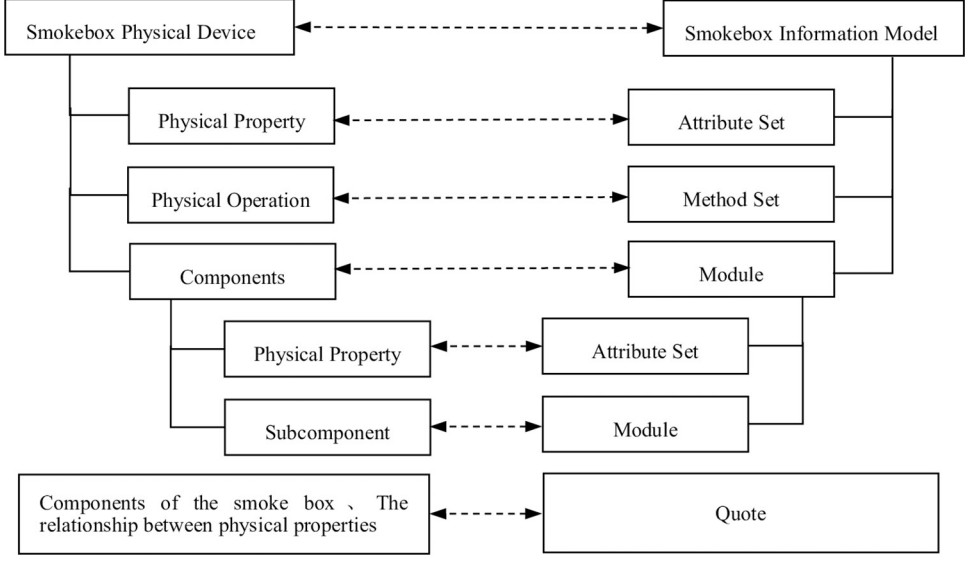

**Fig 8. Mapping between the smoke box and the information model element.**

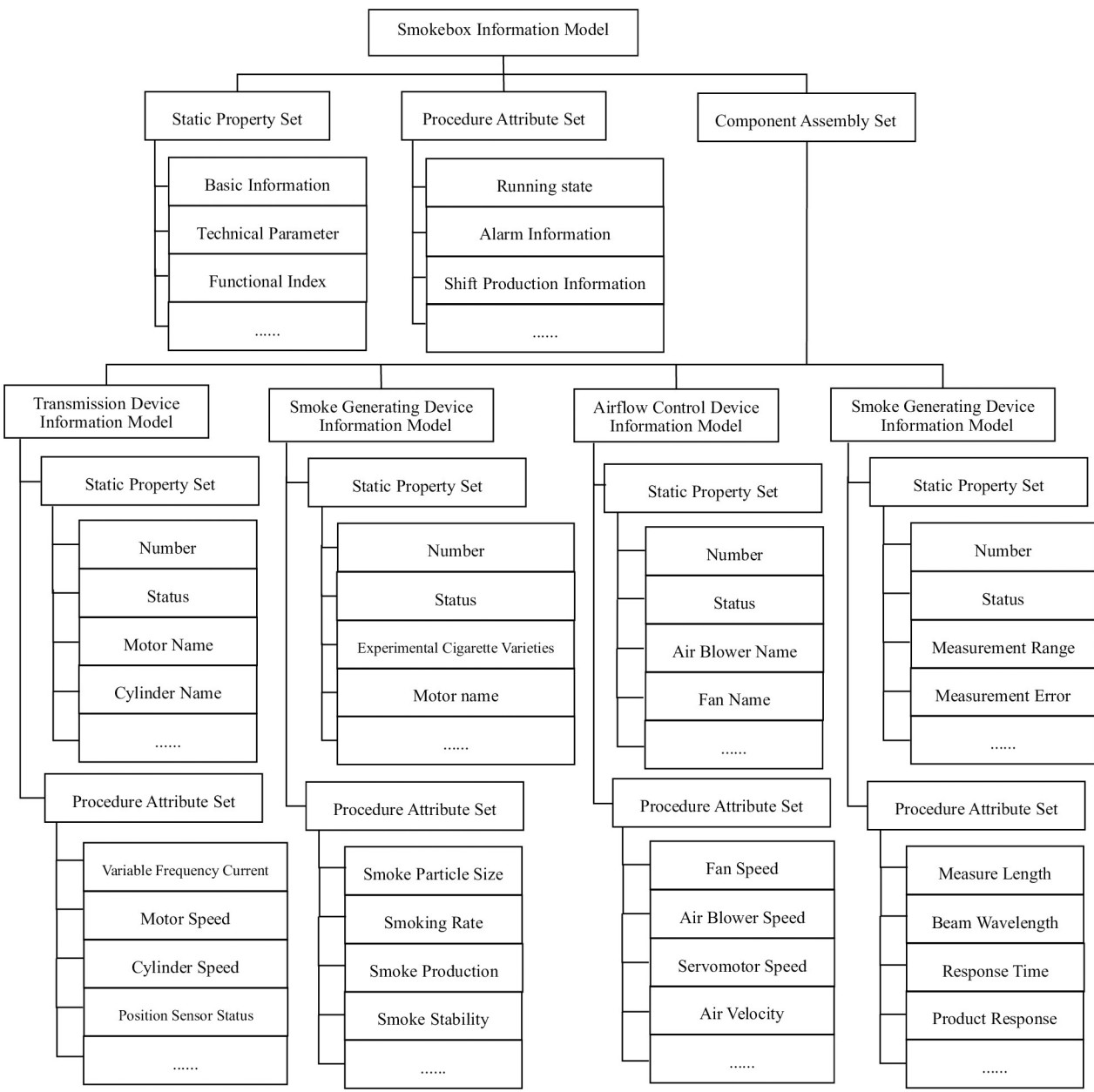

**Fig 9. Information model of smoke box.**

attribute set of the smoke box information model mainly includes the basic information of the smoke box (box name, model, number, manufacturer, specifications, etc.), technical parameters (airflow speed, smoke generation rate, input voltage, measurement error, humidity, temperature, etc.), functional indicators (response threshold calibration, response threshold experiment, consistency experiment, orientation experiment, repeatability experiment, etc.), and process attributes include operation information (calibration alarm name, calibration alarm model, calibration alarm order number, operation technical parameters, etc.), alarm information (unqualified alarm name, unqualified alarm model, unqualified alarm order number, environmental parameter overrun alarm information, equipment failure alarm, etc.), and

shift production information (shift number, personnel on duty, production time, downtime). The static attribute set of the transmission device information model includes the number, state, motor name, cylinder name, etc., and the process attributes include frequency conversion current, motor speed, cylinder speed, position sensor status, product position, etc. The static attribute set of the smoke generating device information model includes the number, state, experimental smoke variety, and the process attributes include smoke particle size, smoke generation rate, smoke production volume, experimental smoke stability, smoke generation control mode, etc. The static attribute set of the airflow regulating device information model includes the number, state, fan name, fan quantity, fan name, fan control mode, electric hanging basket name, electric hanging basket control. The process attributes include fan speed, fan speed, servo motor speed, hanging basket azimuth angle, airflow speed, temperature and humidity, shielding rate. The static attribute set of the measuring device information model includes the number, state, measurement range, measurement error, resolution ratio, light source, receiver, etc., and the process attributes include optical measurement length, beam wavelength, response time, product response, light sensor data.

## 4.2 Address space mapping

The description file of the smoke box is described using standard XML syntax. The information model loader parses the input XML file, maps different information model elements to the OPC UA model, and automatically generates address spaces. The mapping between smoke box information and OPC UA includes structural data mapping and pure data mapping. Structural data is for organizational structure relationships, not actual data. This type of structural information can be represented by the OPC UA "FloderType" object type. Pure data refers to the information represented by attributes. The mapping of the OPC UA model is determined based on different data types and included attribute elements. Generally, the state quantity of two values is represented by "TwoDiscrteStateType", multiple state values are represented by "MultiStateType", analog quantities are represented by "AnalogItemType", and others are represented by "DataItemType". The mapping between smoke box information models is shown in Table 1 (due to the large amount of content, only a portion is selected).

## 4.3 Address space management

The address space mainly implements corresponding read, write, and subscribe operations based on the established mapping table, as shown in Fig 10. When the memory data point changes, it is updated to the OPC UA address space. When the OPC UA client reads node data, the corresponding node data is directly returned from the address space. When the OPC UA client writes data, the system provides a mechanism to ensure that the update of the memory point is associated with the actual device IO. When the OPC UA client subscribes to a node, the system can provide a mechanism to maintain the correspondence between the information of the memory point and the change of the node value in the OPC UA address space.

## 4.4 Application of information model

This article uses Siemens NX platform to construct a digital twin model for the smoke box, builds a virtual-reality communication platform using OPC UA communication protocol, and finally uses real-time mapping technology to realize 3D model visualization and virtual production visualization. Siemens NX is equipped with an OPC UA client, which can realize data exchange between the digital twin model's data, signals, and variables and external servers. The entire environment consists of three main components: PLC CPU, OPC server and digital twin model (MCD). The CPU is used to process the logic of operation, obtain the machine

**Table 1. Mapping of the information model of the smoke chamber.**

| Number | Information Model Element Point | OPC UA Meta-Model Types | Citation Relationship | Remarks |
|---|---|---|---|---|
| 1 | Smoke box | FloderType | Reference under the root node | |
| 2 | Static attributes | FloderType | There is an organizational reference under the smoke box node | |
| 3 | Basic Information | DataItemType | There are attribute references under the static attribute node | |
| 4 | Technical Parameter | DataItemType | There are attribute references under the static attribute node | |
| 5 | Functional Index | DataItemType | There are attribute references under the static attribute node | |
| 6 | Procedure Attribute Set | FloderType | There is an organizational reference under the smoke box node | |
| 7 | Running State | MultiStateType | There are attribute references under the process attribute node | This can use multiple states, such as running, stopping, and failure. |
| 8 | Alarm Information | TwoDiscrteStateType | There are attribute references under the process attribute node | |
| 9 | Shift Production Information | AnalogItemType | There are attribute references under the process attribute node | Since this value is writable, it can be processed accordingly according to the analog output |
| 10 | Component Assembly Set | FloderType | There is an organized reference under the smoke box node | |
| 11 | Transmission Device Information Model | FloderType | There are organizational references under the component node | |
| 12 | Procedure Attribute Set | FloderType | Organized references under the transmission equipment information model node | |
| 13 | Variable Frequency Current | AnalogItemType | There are organized references under the process attributes of the transmission equipment information model node | Because this value is read-only, it can be processed accordingly according to the analog output |
| 14 | Motor Speed | AnalogItemType | Organized references are available under the process attributes of the transmission equipment information model node | Because this value is read-only, it can be processed accordingly according to the analog output |
| 15 | Cylinder Speed | AnalogItemType | Organized references under the transmission equipment information model process attribute node | |
| 16 | Position Sensor Status | TwoDiscrteStateType | There are organized references under the process attributes of the transmission equipment information model node | |
| 17 | . . .. . .. | | | |

state through the OPC server, and send operating instructions to the device through the OPC server; MCD is used to display the virtual model and perform simulation, obtain CPU instructions from OPC, and feedback the current device state through the OPC server. The data

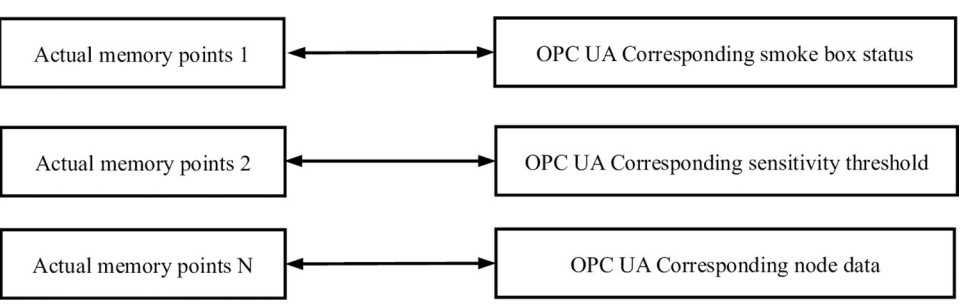

**Fig 10. Address space management.**

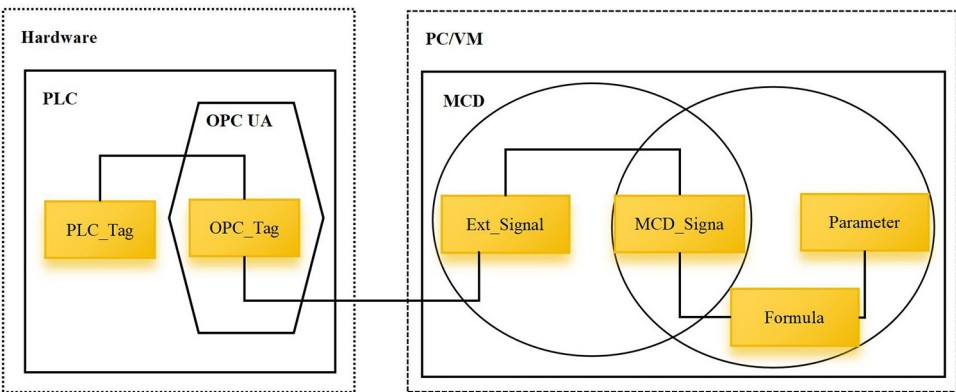

**Fig 11. Data interaction between digital twin model and physical equipment.**

interaction between the digital twin model and physical equipment is shown in Fig 11 [3], and Fig 12 is a comparison diagram of the digital twin model of the smoke box and the physical object.

In the application experiment, a specific model of smoke alarm was chosen to follow its complete lifecycle workflow at the calibration site, spanning from order issuance, production scheduling, product dispatch, calibration procedure, calibration execution, quality inspection, to finished product warehousing. By connecting to the backend data server via Web Service, the updates and modifications in the respective data tables were monitored, thereby verifying the synchronized updating of the information organization within the information model. The detailed verification process is depicted in Fig 13.

The production process information management system of the workshop is illustrated in Fig 14. During the calibration process, the digital twin system displays the real-time operational data and the data change curves for various parameters of the smoke detector. The calibration data for the smoke detector is saved in the database.

The application of digital twin smoke alarm workshop has broken the traditional manual management mode of low efficiency, error-prone, manual record and paper transmission, and solved the drawbacks such as no display of field detection equipment information, no statistics of detection information, no traceability of quality problems, and delay of manual scheduling production information transmission. It realizes the visualization of field equipment information, information statistics, on-time production scheduling, traceability of quality problems,

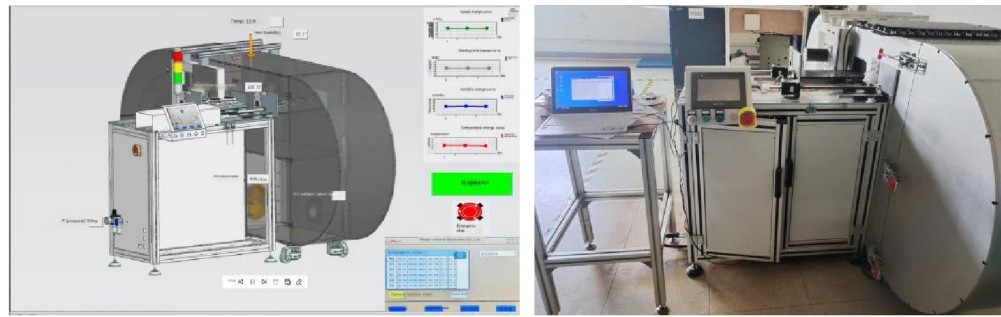

(a) Digital twin model of cigarette box (b)Real cigarette box

**Fig 12. Comparison between the digital twin model of the smoke chamber and the physical object.**

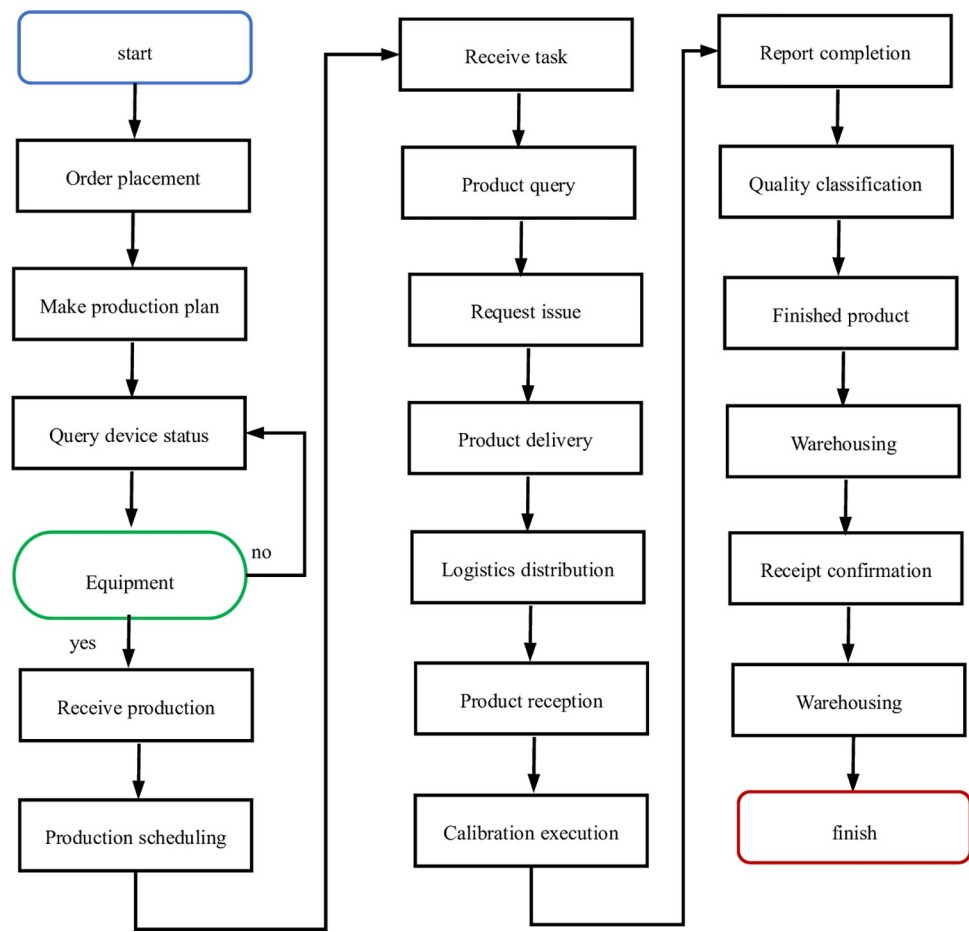

**Fig 13. Flow chart of field application test.**

paperless process management, networked data transmission, and intelligent field management. After the digital twin workshop was put into operation, the production efficiency was increased by 200 percent, and the pass rate was increased from 98 percent to 99.6 percent. The annual production capacity of a single unit was increased from 80W to 160W, and the overall

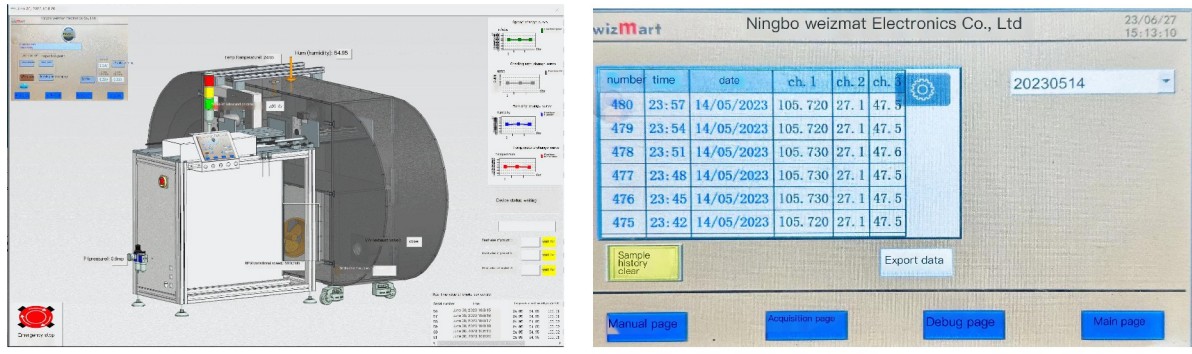

(a)      Twin system monitoring interface             (b)Data acquisition page

**Fig 14. Production process information management system.**

production capacity of 10 units was increased from 800W to 1600W. The repair rate of defective products was reduced by 1.6%, indirectly recovering the economic loss of 51.2W.

## 5 Conclusion

As a key technology for the digital development of enterprises, digital twin technology is the key to information exchange between the physical world and the virtual world. This article applies OPC UA technology to build a virtual-real information interaction platform in the digital twin workshop. Based on the unified architecture of OPC UA, an information model for the digital workshop is proposed. Based on the operation mode of the digital twin workshop and the requirements of virtual-real fusion, an OPC UA-based digital twin smoke alarm calibration workshop data acquisition system architecture is proposed, providing an architectural foundation for multi-source heterogeneous data acquisition in the smoke alarm calibration workshop. The information model of the smoke alarm calibration workshop is constructed from three dimensions: static attribute set, process attribute set, and functional component set, and applied to the calibration digital prototype workshop of Ningbo Weizmat Co., Ltd. Four subcomponent information models of transmission device, smoke generating device, airflow regulating device, and measuring device are constructed and analyzed in detail. After instantiating the information model, the feasibility of the model is verified using an OPC UA server, and data acquisition and transmission are performed with a digital twin platform to achieve the implementation mapping between the twin world and the physical world. This study provides strong technical support for realizing visualization, intelligence, and flexibility in smoke alarm calibration, and provides technical support and reference for realizing intelligent production workshops for smoke alarm production.

## Author Contributions

**Data curation:** Wenfeng Ying.

**Writing – original draft:** Min Wu.

**Writing – review & editing:** Min Wu.

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
