## [Decision Letter · Decision Letter 0]

11 Jun 2024

PONE-D-24-16301Research and application of OPC UA-based digital twin smoke alarm calibration workshop information modelPLOS ONE

Dear Dr. wu,

Thank you for submitting your manuscript to PLOS ONE. After careful consideration, we feel that it has merit but does not fully meet PLOS ONE’s publication criteria as it currently stands. Therefore, we invite you to submit a revised version of the manuscript that addresses the points raised during the review process.

**ACADEMIC EDITOR: Major Revision.**==============================

We look forward to receiving your revised manuscript.

Kind regards,

Worradorn Phairuang, Ph.D.

Academic Editor

PLOS ONE

“This work was supported by 2022 Visiting Engineer of Colleges and Universities in Zhejiang Province(Project No. FG2022037).”

Reviewers' comments:

Reviewer's Responses to Questions

**Comments to the Author**

1. Is the manuscript technically sound, and do the data support the conclusions?

Reviewer #1: Partly

Reviewer #2: Yes

2. Has the statistical analysis been performed appropriately and rigorously? 

Reviewer #1: I Don't Know

Reviewer #2: Yes

3. Have the authors made all data underlying the findings in their manuscript fully available?

Reviewer #1: Yes

Reviewer #2: Yes

4. Is the manuscript presented in an intelligible fashion and written in standard English?

Reviewer #1: Yes

Reviewer #2: Yes

5. Review Comments to the Author

Reviewer #1: The paper proposes an innovative framework integrating OPC UA and digital twin technology to enhance data integration and information sharing in smoke alarm calibration workshops. The paper needs to address the following issues:

1. When abbreviations first appear in the text, the full term should be provided to facilitate reader understanding.

2. The review of the application of OPC UA in the field of smoke alarms is not comprehensive enough and should be supplemented with the current status of relevant technologies in this field.

3. Conceptual introductions in the text should be accompanied by references.

4. Section 3.1 claims that a "digital twin smoke alarm calibration workshop information model" was established; the basis for the architecture's establishment should be described in detail.

5. The author states, "Through testing and verification of the digital twin system, the calibration process of the smoke box is visually monitored." The effects of the digital twin system should be specifically demonstrated.

6. Chinese labels in the images should be provided with English annotations.

Overall, the authors should provide a detailed introduction to the validation effects of the digital twin system and supplement it with relevant results.

Reviewer #2: This paper applies OPC UA technology to build a virtual-real information interaction platform in the digital twin workshop, which provides strong technical support for realizing visualization, intelligence, and flexibility in smoke alarm calibration, and provides technical support and reference for realizing intelligent production workshops for smoke alarm production. It has applied to the calibration digital prototype workshop of Ningbo Weizmat Co., Ltd. Therefore, I think this research is very meaningful and has great practical value. However, I suggest that this manuscript still needs some minor revisions as follows:

1.It is best to label and have a title below the figures appearing in the manuscript for easy reference by readers.

2.The entries in the references should have a unified format.

3.Some minor errors, including spelling, grammar, punctuation, etc., have been marked in the manuscript as attached.

6. PLOS authors have the option to publish the peer review history of their article (what does this mean?). If published, this will include your full peer review and any attached files.

Reviewer #1: No

Reviewer #2: No

---

## [Author Response · Author response to Decision Letter 0]

18 Jun 2024

We thank you very much for giving us another chance to revise and improve our manuscript. We try our best to revise the paper according to the reviewers’ suggestions, We have made changes to the proposed suggestions one by one and uploaded them as documents.

We hope the revised paper has addressed all the reviewers’ concerns and reached the requirements of the journal for publishing.

---

## [Decision Letter · Decision Letter 1]

12 Jul 2024

Research and application of OPC UA-based digital twin smoke alarm calibration workshop information model

PONE-D-24-16301R1

Dear Dr. wu,

We’re pleased to inform you that your manuscript has been judged scientifically suitable for publication and will be formally accepted for publication once it meets all outstanding technical requirements.

Kind regards,

Worradorn Phairuang, Ph.D.

Academic Editor

PLOS ONE

Additional Editor Comments (optional):

Reviewers' comments:

Reviewer's Responses to Questions

**Comments to the Author**

1. If the authors have adequately addressed your comments raised in a previous round of review and you feel that this manuscript is now acceptable for publication, you may indicate that here to bypass the “Comments to the Author” section, enter your conflict of interest statement in the “Confidential to Editor” section, and submit your "Accept" recommendation.

Reviewer #1: All comments have been addressed

Reviewer #2: All comments have been addressed

2. Is the manuscript technically sound, and do the data support the conclusions?

Reviewer #1: Yes

Reviewer #2: Yes

3. Has the statistical analysis been performed appropriately and rigorously? 

Reviewer #1: Yes

Reviewer #2: Yes

4. Have the authors made all data underlying the findings in their manuscript fully available?

Reviewer #1: Yes

Reviewer #2: Yes

5. Is the manuscript presented in an intelligible fashion and written in standard English?

Reviewer #1: Yes

Reviewer #2: Yes

6. Review Comments to the Author

Reviewer #1: (No Response)

Reviewer #2: The authors have made very good modifications based on the comments of the reviewers. Based on this, I suggest that this paper can be accepted in this journal.

7. PLOS authors have the option to publish the peer review history of their article (what does this mean?). If published, this will include your full peer review and any attached files.

Reviewer #1: No

Reviewer #2: No

---

## [Editor Report · Acceptance letter]

15 Jul 2024

PONE-D-24-16301R1 

PLOS ONE

Dear Dr. Wu, 

I'm pleased to inform you that your manuscript has been deemed suitable for publication in PLOS ONE. Congratulations! Your manuscript is now being handed over to our production team.

Kind regards, 

on behalf of

Dr. Worradorn Phairuang 

Academic Editor

PLOS ONE